# Hepatitis Delta Virus and Hepatocellular Carcinoma

**DOI:** 10.3390/pathogens13050362

**Published:** 2024-04-27

**Authors:** Daniele Lombardo, Maria Stella Franzè, Giuseppe Caminiti, Teresa Pollicino

**Affiliations:** Department of Clinical and Experimental Medicine, University Hospital of Messina, 98124 Messina, Italy; daniele.lombardo@unime.it (D.L.); mariastellafranze@gmail.com (M.S.F.); giuseppecaminiti8@gmail.com (G.C.)

**Keywords:** HDV, HBV, HCC, cirrhosis, chronic hepatitis, liver disease

## Abstract

The hepatitis D virus (HDV) is a compact, enveloped, circular RNA virus that relies on hepatitis B virus (HBV) envelope proteins to initiate a primary infection in hepatocytes, assemble, and secrete new virions. Globally, HDV infection affects an estimated 12 million to 72 million people, carrying a significantly elevated risk of developing cirrhosis, liver failure, and hepatocellular carcinoma (HCC) compared to an HBV mono-infection. Furthermore, HDV-associated HCC often manifests at a younger age and exhibits more aggressive characteristics. The intricate mechanisms driving the synergistic carcinogenicity of the HDV and HBV are not fully elucidated but are believed to involve chronic inflammation, immune dysregulation, and the direct oncogenic effects of the HDV. Indeed, recent data highlight that the molecular profile of HCC associated with HDV is unique and distinct from that of HBV-induced HCC. However, the question of whether the HDV is an oncogenic virus remains unanswered. In this review, we comprehensively examined several crucial aspects of the HDV, encompassing its epidemiology, molecular biology, immunology, and the associated risks of liver disease progression and HCC development.

## 1. Introduction

The hepatitis delta virus (HDV) is the causative agent of the most severe form of viral hepatitis, and is associated with a rapid progression to cirrhosis and the high rate of development of hepatocellular carcinoma (HCC), a primary malignancy of the liver [1,2]. HDV is the smallest known human virus and is a satellite virus of the hepatitis B virus (HBV), relying on envelope proteins of the HBV to maintain its productive infection [3,4]. The World Health Organization (WHO) estimates that approximately 296 million people are chronically infected with HBV globally, and that about 5% of HBV-positive individuals are co-infected with HDV [5,6,7,8]. However, HDV prevalence may be underestimated because of inadequate testing in individuals who test positive for the HBV surface antigen (HBsAg) (especially those with advanced liver disease). This underestimation is further compounded by the inconsistent performance of serological tests and the variability in sampling methods [5,9,10]. Consequently, the exact global prevalence of HDV infection remains uncertain. Nonetheless, important evidence exists that indicates a significantly higher risk of HCC development in patients with chronic hepatitis D (CHD) compared to those infected only with the HBV [11,12,13,14,15]. This association is further supported by a recent systematic review, which found an increased risk of HCC development in individuals co-infected with the HBV and HDV, with an odds ratio (OR) of 1.28 compared to those with an HBV infection only. Notably, this elevated risk was even more pronounced in prospective cohort studies, where the OR was found to be 2.77 [12]. The development of HCC in the context of HDV infection is particularly concerning. HCC is the most common type of primary liver cancer and is characterized by a poor prognosis and high mortality rate [16]. The association between chronic viral hepatitis and HCC is well established, with HBV and hepatitis C virus (HCV) being among the most common etiological agents [17,18,19]. However, the role of HDV in the carcinogenic process is less clear and represents a critical area of investigation. Several hypotheses have been proposed, including the role of chronic inflammation, hepatic regeneration, and the direct oncogenic potential of HDV [2,20,21].

In this manuscript, we will explore the current understanding of the HDV’s biology, its interaction with HBV, and the pathophysiological mechanisms leading to liver damage. Our manuscript will undertake a thorough review of the evidence linking HDV infection to HCC, examining both clinical studies and molecular research that shed light on this association. The manuscript aims to provide a comprehensive overview of the current state of knowledge in this area, identify gaps in our understanding, and suggest future research directions.

## 2. Virological Aspects

In the late 1970s, Rizzetto and his team in Turin, Italy, identified the delta antigen, a novel immunological marker found in severe HBV infections. Initially thought to be a variant of the HBV, subsequent research revealed distinct features, including a genome significantly smaller than that of the HBV and other animal viruses [22,23]. The use of chimpanzee models at the National Institutes of Health (NIH) in 1978 was instrumental in the isolation of HDV particles. These particles, approximately 36 nm in diameter, contained a 1.7 kb RNA genome, leading to the formal identification of the HDV in 1980 as the only member of the genus Deltavirus in the Deltaviridae family [23,24]. More recently, the HDV and similar viruses have been reclassified into the Kolmioviridae family, currently the only family of the Ribozyviria realm [25].

Eight distinct HDV genotypes have been recognized, with genetic sequence similarities ranging between 81% and 89% [26].

Genotype 1, the most prevalent, is found in Europe, North America, and Mongolia, where HDV infection is highly endemic. Genotype 2 is prevalent in the Far East, while genotype 3 is restricted to the Amazon basin. Following the reclassification of previously known genotype 2b as genotype 4, newly identified HDV genotypes have been classified as 5 to 8, with African sequences falling into these genotypes [26]. The HDV virion consists of a ribonucleoprotein (RNP) core complex and an HBV-encoded envelope. The RNP complex contains a single-stranded, covalently closed circular RNA molecule with a negative polarity, consisting of 1672 to 1697 ribonucleotides, depending on the genotype, and two isoforms of the HDAg: small-HDAg (S-HDAg) of 24 KDa and large-HDAg (L-HDAg) of 27 KDa [4,27,28,29]. The envelope consists of an endoplasmic reticulum (ER)-derived lipid bilayer that embeds the three HBV envelope proteins: large (L-), medium (M-), and small (S-) HBsAgs (Figure 1) [4,27,28,29].

The two isoforms of the HDAg are essentially identical except for an additional 19–20 amino acid residues (genotype dependent) at the C-terminus of the L-HDAg. Despite their structural similarity, they have different biological functions. The S-HDAg is expressed early during infection and is required to initiate and promote HDV replication. In contrast, the L-HDAg, which results from an RNA editing event triggered by the host adenine deaminase (the ADAR-1 enzyme), inhibits viral replication [4,27,28,29,30]. This inhibition promotes the packaging of mature virions and their secretion through its interaction with the self-assembly competent HBV envelope proteins. The packaging process is made possible by the addition of an isoprenoid prosthetic group (prenylation) to the C-terminus of the L-HDAg, a reaction facilitated by the cellular enzyme farnesyl transferase [4,27,28,29].

HDV replication generates three distinct RNA species: a genomic, complementary antigenomic RNA of positive polarity, and a 0.8 kb messenger RNA (mRNA) of antigenomic polarity; the mRNA contains an open reading frame for the translation of HDAgs [4,27,28,29]. The genomic and antigenomic RNAs contain a domain of approximately 100 nucleotides that acts as a ribozyme, cleaving the viral RNA at specific sites without the participation of viral encoded enzymes [31]. Both genomic and antigenomic RNAs are characterized by a high degree of intramolecular base pairing (~74%), resulting in the formation of recurrent back-folded stretches of base-paired rods interspersed with short loop regions [4,27,28,32,33] (Figure 1).

Due to their shared envelope, the HDV and HBV show a marked tropism to human hepatocytes [4,27,28,29,34,35]. The HDV utilizes the same cell receptor as the HBV, the sodium taurocholate co-transporting polypeptide (NTCP), for entry into hepatocytes. This interaction occurs through the pre-S1 domain of the L-HBsAg isoform. It is hypothesized that the processes of internalization and membrane fusion during HDV entry parallel those observed in HBV [29,34,35]. However, direct comparative analyses of both viruses are lacking.

Once the HDV RNP is released into the cytoplasm of hepatocytes, the subsequent steps of HDV replication occur independently of the HBV. This HDV RNP is then transported to the cell nucleus, where RNA replication begins [36,37] (Figure 2).

The incoming genome acts as a template for the initial round of rolling circle amplification (RCA). This process produces linear, multimeric antigenomic RNAs. These RNAs undergo self-cleavage, facilitated by the inherent antigenomic ribozyme, and are subsequently ligated to form circular antigenomic monomers. In a similar fashion, genomes are generated through a second round of RCA, using the newly created antigenomes as templates. This is further refined by the genomic ribozyme. Diverging from other negative-strand RNA viruses, the HDV does not produce its own RNA-dependent RNA polymerase (RdRP). Rather, it co-opts the host cell’s DNA-dependent RNA polymerases (Pols) for the purpose of RNA replication. Compelling evidence supports that RNA Pol-II plays a key role in synthesizing both the HDV genome and its mRNA [4,27,28,29,34,35,38,39].

In cells where the HBsAg is co-expressed, the prenylated L-HDAg recognizes a hydrophobic component in the cytosolic loop of the S-HBsAg. Given that the expression of an S-HBsAg may initiate the self-assembly and release of HBV subviral particles (SVPs), the presence of HBsAgs in cells containing the HDV RNP is sufficient to facilitate HDV secretion. Indeed, through the incorporation of an L-HBsAg, the particles gain infectivity and support transmission into NTCP-receptor expressing cells, enabling the HDV to spread both within the liver and to other hosts [29,40].

The extracellular spread of HDV can be inhibited by Bulevirtide (BLV), an HDV entry inhibitor that was granted full marketing authorization by the European Medicines Agency for the treatment of CHD and compensated liver disease in April 2023 [41,42]. Additionally, the investigational drug Lonafarnib, which acts as an HDV secretion inhibitor, can indirectly block this spread. Lonafarnib works by inhibiting the prenylation of the L-HDAg, thereby preventing the acquisition of the HBV envelope by the HDV [43,44,45].

While inhibiting the extracellular spread of the HDV significantly reduces its activity and propagation in cell culture models, animals, and patients, emerging evidence suggests an alternative mechanism of HDV infection persistence that does not rely on new cell entry. This is indicated by two key observations: firstly, HDV-positive hepatocytes were found in patients more than a year after liver transplantation, even in the absence of HBV DNA and serum-HBsAgs [46,47,48,49]; secondly, an HDV mono-infection continued for at least six weeks in humanized mice lacking the HBV, and this infection could be reactivated by superinfection with the HBV [50]. Furthermore, a recent study identified an additional pathway for the spread of the HDV in cell lines and in mice that had been transplanted with primary human hepatocytes (PHHs) [50]: the spread of the HDV through cell division.

## 3. Epidemiology

Over the past three decades, the prevalence of HDV infections has changed, mirroring the impact of global HBV vaccination programs in controlling HBV infections. These vaccination programs have been instrumental in decreasing the population of HBsAg carriers, who are at risk of HDV infection. Consequently, as a secondary benefit, these programs have played a significant role in the worldwide reduction of HDV infections [35,51]. Although HDV infections occur worldwide, the exact prevalence in many countries remains to be determined. HDV prevalence is less than 1% in the general population of North America and Northern Europe, but exceeds 20% in certain geographical areas such as Mauritania, Gabon, Benin, Cameroon, Senegal, Iran, Peru, the western Brazilian Amazon, Venezuela, Romania, Pakistan, Tajikistan, and Mongolia [5,9,10,52,53,54].

For individuals who test positive for the HBsAg, the prevalence of HDV infection shows significant variation. In some Western countries, it is less than 2%, whereas in HBsAg-positive patients from certain low- and middle-income countries, specifically in sub-Saharan Africa, India, Mongolia, and Brazil, where HBV is endemic, the prevalence ranges from 3% to 80% [1,5,9,10,52,53,54,55].

In the United States, the prevalence of HDV antibodies (anti-HDV) among HBsAg-positive adults has been reported to range from as low as 3.4% to 6%, and in some instances, up to 43%. Furthermore, the prevalence of HDV infection also varies significantly among different risk groups, with a prevalence of 37% and 17% among injecting drug users and people with high-risk sexual behaviors, respectively.

Three recent, large-scale meta-analyses have shown different prevalence rates of HDV infection: 0.11% to 0.98% in the general population, 4.5% to 13.02% in HBsAg-positive carriers, and 14.6% to 18.6% in patients attending hepatology clinics [5,9,10].

These rates suggest that there are approximately 12 to 72 million people worldwide with serological evidence of HDV exposure [5,9,10]. The variation in the reported HDV infection rates can be attributed to several factors. A major reason for this is the lack of comprehensive population-based studies combined with different screening strategies from different countries and recommended by different scientific societies. For instance, while the European Association for the Study of the Liver (EASL) and the Asian Pacific Association for the Study of the Liver (APASL) recommend testing all HBsAg-positive samples for anti-HDV antibodies, the American Association for the Study of Liver Diseases (AASLD) advises testing only high-risk patients [1,56,57,58,59]. Additionally, testing approaches are not standardized: not all positive anti-HDV antibody tests are followed up with an HDV RNA assessment, and HDV RNA quantification lacks standardization, leading to variability both within and between assays. Finally, incomplete data from various regions contribute to gaps in the global understanding of HDV epidemiology [59].

## 4. Clinical Outcomes of HDV Infection

Due to its obligate dependence on the HBV for replication, an HDV infection occurs in two primary forms: simultaneous infection or co-infection with both the HBV and HDV, and HDV superinfection in a chronic HBV carrier [35,60,61]. A third, less common condition is the HBV-independent HDV mono-infection, which is typically non-productive and can be rescued upon a subsequent HBV infection. This condition has been reported in several cases following liver transplantation [47,49,62,63].

HBV and HDV co-infection usually lead to acute hepatitis, with symptoms indistinguishable from those of a typical acute HBV infection. The severity of this condition can vary widely, ranging from mild to severe, and is usually followed by clearance of both viruses. However, a minority of co-infected patients may progress to acute liver failure. The risk of acute liver failure is much higher than during an acute HBV mono-infection [35,60,61,62,64]. An acute HBV–HDV co-infection may exhibit either a single or a double increase in liver enzyme levels, commonly observed as two distinct peaks separated by approximately 2–5 weeks. This characteristic pattern, termed biphasic hepatitis, entails a recurrence of elevated ALT and AST levels following a temporary amelioration. The presumed mechanism behind this phenomenon is the sequential replication of the HBV followed by the HDV [65]. A small proportion of patients with an HDV and HBV co-infection (less than 5%) progress to chronic infection and are at risk of developing cirrhosis and hepatocellular carcinoma (HCC) [35,60,61].

An HDV superinfection is commonly associated with an episode of acute hepatitis that, in more than 90% of superinfected HBV carriers, progresses to a chronic dual infection. In this setting, the risk of acute liver failure is particularly high. Once a chronic HBV/HDV co-infection is established, preexisting liver disease worsens [35,60,61]. In this context, HBV replication is typically (but not always) suppressed, as demonstrated in several in vivo and in vitro studies [66,67,68,69]. Nevertheless, chronic hepatitis D has an accelerated progression, leading more rapidly to cirrhosis and exhibiting a higher incidence of liver-related mortality and hepatocellular carcinoma compared to chronic hepatitis B infection alone [1,35,60,61].

Persistent HDV viremia is the most important risk factor associated with progression to cirrhosis and mortality [35,61,70,71,72,73].

Due to the paucity of data on virus genotype characterization, it is not possible to assess the impact of HDV and HBV genotypes on CHD progression. However, there is evidence indicating that virus genotypes may associate with HDV and HBV infection-specific clinical outcomes [71,74,75,76]. Studies from Taiwan have shown that an HDV genotype 1 infection is associated with a more severe clinical course than HDV genotype 2. Similar findings have been observed for HBV genotype C compared with HBV genotype B [77]. In addition, among the different HDV genotypes, genotype 3—mostly found in South America—has been associated with a severe form of hepatitis [78], while genotype 5 has often been associated with slower disease progression and a favorable response to interferon-alpha (IFNα) [79]. However, this latter observation has been challenged by a recent French nationwide retrospective study on 1112 anti-HDV-positive patients, showing that both European genotype 1 and African genotype 5 were associated with a higher risk of developing cirrhosis. Interestingly, however, patients from sub-Saharan Africa had a lower incidence of cirrhosis than their European counterparts [71]. This finding suggests that ethnicity, independent of genotype, may play an important role in determining the progression of CHD. However, more research is needed to better understand the impact of both HDV and HBV genotypes beyond ethnic factors.

Other factors associated with an increased risk of liver disease progression include high serum HBV-DNA levels [80,81,82], alcohol use (>2 drinks/d for men and >1 drink/d for women), obesity, diabetes [1,61,83], and concomitant HIV infection [1,84].

## 5. HDV Pathogenesis

### Immunological Aspects

An HDV infection triggers immune-mediated liver damage, which plays a critical role in HDV pathogenesis. Nevertheless, the intricacies of the HDV’s interaction with the immune system remain partially veiled. Unlike its partner, HBV, the HDV stimulates the production of interferons, pro-inflammatory cytokines, and interferon-stimulated genes (ISGs) both in vitro [85,86,87,88,89,90,91,92] and in animal models [90,93,94,95,96]. This response, though able to suppress the HBV, appears insufficient to effectively control HDV replication in cells and humanized mice [90,91,93], suggesting further complexities at play. Noteworthy, most studies investigating the HDV’s interaction with the innate immune system are based on acute infections with a high viral replication. Whether these findings can be extended to chronic infections remains unclear. In addition, the significant variability in the HDV viral load and replication levels among patients suggests that the strength of the innate immune response may vary accordingly. This highlights the need for further research to explore these complexities and their impact on disease progression.

Natural killer (NK) cells are innate effector cells that are abundantly present in liver tissue and are known for their anti-viral activity [97,98]. A higher frequency of NK cells with an unaltered phenotypic differentiation status prior to treatment with IFN-alpha was positively associated with a therapeutic reduction in serum HDV RNA [99], highlighting the importance of NK cell activity in an HDV infection. Recent evidence has also shown that activated natural killer (NK) cells have the potential to eliminate HDV-infected cells through the tumor necrosis factor-related apoptosis-inducing ligand (TRAIL)/TRAIL receptor 2 (R2) pathway [100]. Although it remains to be evaluated whether NK cells may also mediate liver damage, these latter data suggest a potential for NK cells as effector cells for HDV clearance.

Mucosal-associated invariant T (MAIT) cells are a unique subset of innate-like T cells found in abundance in the human liver, gut mucosa, and other tissues [101]. They play a vital role in defending against bacterial and fungal infections by utilizing the riboflavin metabolic pathway. Upon recognizing microbial-derived riboflavin metabolites presented by the major histocompatibility complex class I-related (MR1) protein, MAIT cells quickly release pro-inflammatory cytokines like interferon-gamma (IFN-γ) and tumor necrosis factor (TNF), as well as other immune molecules like IL-17 and IL-22 (which help recruit immune cells and activate inflammatory responses). They can also directly kill infected cells and secrete antimicrobial molecules. Interestingly, MAIT cells can also be activated independently of MR1 by cytokines like IL-12 and IL-18, which are secreted by antigen-presenting cells (APCs) [101].

Dias J. et al. [102] analyzed MAIT cells from the peripheral blood of 41 patients with CHD, 38 patients with an HBV mono-infection, and 57 healthy controls. Furthermore, they examined liver biopsies from three patients with an HDV infection and from seven HDV- and HBV-negative control cases. Patients with a chronic HDV infection showed a dramatic decrease in both circulating and liver-resident MAIT cells compared to patients with an HBV mono-infection. This depletion was coupled with functional impairment in the remaining MAIT cell population. Elevated levels of IL-12 and IL-18, pro-inflammatory cytokines linked to MAIT cell death and monocyte activation, were observed in HDV patients [102]. This suggests a three-stage process characterized by initial MAIT cell activation upon HDV infection, followed by functional impairment, and ultimately, depletion.

The innate immune response is activated during an HDV infection, while the adaptive immune response is typically weak and fails to consistently clear the virus. Mimicking the cunning tactics of the HBV, the HDV thwarts immune attacks driven by IFN-α, enabling both its persistence and the survival of infected cells. This strategy creates a favorable environment for virus growth [103]. In cases of acute-resolving HDV infection, anti-HDV antibodies (Abs) are found in relatively low titers, whereas in persistent infections, anti-HDV Abs are detectable at higher titers [104]. In patients with chronic active hepatitis, anti-HDV IgM often remains at high levels. This pattern suggests that anti-HDV Abs play a minimal role in controlling and clearing the virus. This is likely due to their lack of neutralizing activity [105].

The role of T cells in an HBV/HDV co-infection remains unclear because of the lack of suitable animal models and the limited identification and fine-mapping of HDV-specific T cell epitopes with defined HLA restrictions. To date, the HDV-specific CD4+ T cell epitope repertoire has been analyzed in detail in few studies [106,107]. Around 30–40% of untreated patients with an HDV infection showed HDV-specific CD4+ T cell responses, typically targeting 1–3 different epitopes. These responses were characterized as weak and were only detectable ex vivo after antigen-specific stimulation. Furthermore, these studies have yielded inconsistent results regarding the association between detectable HDV-specific CD4+ T cell responses and clinical parameters [106,107].

Recent studies have thoroughly investigated HDV-specific CD8+ T cell responses [107,108,109]. It has been shown that this response is not present in all CHD patients. Indeed, approximately 40% of untreated patients and 70% of patients treated with Lonafarnib exhibit such a response. This was determined after the ex vivo stimulation of CD8+ T cells with overlapping peptides derived from the L-HDAg [107,109]. The CD8+ T cell epitopes specific to the HDV were primarily restricted by HLA-B alleles and mainly located in the C-terminal region of the HDAg, which is exclusive to its large isoform [107,109]. Thus, similar to HDV-specific CD4+ T cells, HDV-specific CD8+ T cells were also distinguished by low ex vivo frequencies [108,109].

Recent studies have indicated that in a chronic HBV/HDV co-infection, CD8+ T cells specific to the HDV may not be effective due to mechanisms of failure that are common in other viral infections, such as mutational viral escape and CD8+ T cell exhaustion [108,109,110,111]. An international study of 104 untreated patients with a chronic HBV/HDV co-infection revealed HLA class I-associated viral sequence polymorphisms, identifying several HDV-specific CD8+ T cell epitopes. Variations in these epitopes were shown to facilitate viral escape [108,111]. Notably, a limited repertoire of HDV-specific CD8+ T cell epitopes was identified, with a significant dominance of HLA-B alleles in mediating HDV-specific responses [107,108,109]. Interestingly, the majority of identified epitopes and associated polymorphisms were linked to rare HLA class I alleles [108], suggesting that common HLA alleles (like HLA-A*02) play a minimal role in the HDV-specific CD8+ T cell response. This suggests that viral escape mechanisms may have eliminated HDV-specific CD8+ T cell epitopes restricted by common HLA class I alleles at the population level.

HDV-specific CD8+ T cells targeting mutated viral epitopes exhibited a memory-like phenotype characterized by the positive expression of CD127, programmed cell death protein 1 (PD-1), and T cell factor 1 (TCF-1), alongside a low expression of activation markers like CD38 [108,109]. In contrast, HDV-specific CD8+ T cells targeting non-mutated epitopes showed a higher expression of CD38 and lower expression of CD127 and TCF-1 [109]. These cells were not terminally exhausted, as evidenced by their low levels of CD57 and notably lower expression of multiple inhibitory/exhaustion markers compared to CD8+ T cells specific to the HBV, CMV, and EBV [109]. At first glance, it may appear surprising that virus-specific CD8+ T cells in severe viral hepatitis exhibit lower signs of exhaustion compared to those in other chronic infections. This observation might shed light on the consistent strength of HDV-specific CD8+ T cell responses in both resolved and ongoing infections. This is in accordance with research indicating that HDV-specific T cells can be effectively reactivated using the cytokine IL-12 instead of checkpoint inhibitors like anti-PD-L1 or anti-CTLA4 [110]. Furthermore, CD38+ HDV-specific CD8+ T cells, which target unmutated HDV epitopes, have been associated with elevated levels of aspartate aminotransferase (AST). This association hints at a potential, albeit not definitively established, contribution of these T cells to the immunopathology observed in a chronic HBV/HDV co-infection [109].

In conclusion, the capacity of the HDV to elude the host’s immune defenses by selecting variants and impairing the functions of immune cells presents significant obstacles in effectively securing control of the infection. This often prolonged and insufficient immune reaction not only struggles to suppress the HDV but might also expedite the progression of liver disease. The immune system’s response to the HDV, characterized by the release of pro-inflammatory cytokines and the immune-driven destruction of hepatocytes, could potentially create an environment conducive to carcinogenesis. Therefore, it is imperative to achieve a deep understanding of the immune response dynamics and develop targeted strategies to eradicate HDV infections. Doing so is essential for addressing the complex challenges posed by an HDV infection and its comprehensive management.

## 6. HDV and HCC Development: Potential Oncogenic Mechanisms

More than 90% of HCC cases occur in the context of chronic liver disease with cirrhosis, regardless of cause. Indeed, cirrhosis the most important risk factor for the development of HCC [16]. The advancements in tumor biology and molecular genetic profiling have paved the way for the identification of a multitude of signaling pathways and molecular mechanisms critical in the initiation and promotion of HCC. These discoveries have significantly enriched our understanding of HCC pathogenesis, opening new avenues for targeted therapies and personalized treatment strategies [16,112]. Despite these insights, data on the potential oncogenic mechanisms of the HDV remain significantly limited.

A previous study using transgenic mice that expressed HDAgs in their hepatocytes suggested that it was not directly cytotoxic to the liver, as no liver damage was observed after 18 months [113]. However, recent research using adeno-associated vectors (AAVs) carrying functional HBV and HDV genomes has challenged this view. These new studies suggest that HDAgs, particularly the S-HDAg form, may contribute to liver injury even without the involvement of immune T cells [114,115]. In support of this notion, it has been observed that the L-HDAg expressed in human liver (Huh7) and kidney (HEK293) cells can disrupt the TNFα-NF-κB signaling pathway [116], which is a key driver of inflammation. It is worth noting that chronic inflammation, which is triggered by TNFα, has been linked to liver damage progression, including fibrosis and cirrhosis [117]. Moreover, an examination of serum samples from patients with a chronic HDV infection has revealed a connection between TNFα levels and HDV RNA loads, indicating a possible involvement of the L-HDAg in liver damage progression [118]. A number of studies have suggested multiple mechanisms by which the HDV may affect signaling pathways associated with hepatocarcinogenesis. These mechanisms include impaired cell growth, epigenetic modifications, the targeted dysregulation of long non-coding RNAs (lncRNAs), changes in the immune response, and proteomic modifications [2,20,21,119,120,121]. The activation of transforming growth factor-β (TGF-β) signaling has been proposed as a mechanism behind the rapid progression of liver disease in patients co-infected with HBV/HDV [122]. TGF-β plays a critical role in numerous cellular functions including cell growth, differentiation, the production and degradation of extracellular matrix (ECM) proteins, and apoptosis [123]. TGF-β is also a pro-fibrogenic cytokine that plays a major role in liver fibrosis and cirrhosis. Furthermore, TGF-β acts as a potent inhibitor of hepatocyte proliferation and has been shown to accelerate hepatocarcinogenesis in transgenic mouse models [122]. Choi et al. [124] have shown that the L-HDAg triggers the activation of TGF-β- and c-Jun-dependent signaling pathways. A critical factor in this activation is the isoprenylation of a cysteine residue at the C-terminal of the L-HDAg, which plays a pivotal role. Furthermore, they demonstrated that the L-HDAg enhances the activation of the hepatitis B virus X protein–mediated TGF- β and AP-1 signaling cascades synergistically. This combined activation subsequently leads to the elevated expression of the plasminogen activator inhibitor (PAI)-1 protein. Collectively, these molecular interactions offer insights into why liver disease progresses more rapidly towards cirrhosis and why there is a higher incidence of HCC development in patients co-infected with HBV/HDV compared to those with an HBV mono-infection [124].

Other in vitro studies have demonstrated that the L-HDAg regulates various cellular functions by activating the nuclear factor kappa B (NF-κB) pathways—via the induction of tumor necrosis factor-alpha (TNF-alpha) [116]—and the signal transducer and activator of transcription-3 (STAT-3) [86]. It has been observed that the L-HDAg has the potential to activate NOX4 gene expression, which, in turn, can lead to the release of reactive oxygen species (ROS), thus inducing oxidative stress [125]. The increase in ROS levels subsequently activates both STAT-3 and NF-κB—key players in various cellular signaling pathways—which govern inflammation, apoptosis, cell proliferation, and tumor development. The study also revealed that the use of antioxidants or calcium inhibitors significantly mitigated the activation processes [125].

The L-HDAg has also been involved in the activation of other potential pro-oncogenic mechanisms, including the stimulation of the JAK-STAT pathway [126], via the activation of STAT-3 [125], and the induction of the c-Fos proto-oncogene [127]. Additionally, evidence suggests that the S-HDAg can bind specifically to the transcript of the glutathione S-transferase P1 (GSTP1) tumor suppressor gene, leading to a significant reduction in GSTP1 protein production. Notably, the transfection of L-02 human fetal hepatocytes with a recombinant vector expressing S-HDAg led to, besides a decrease in GSTP1 production, a significant accumulation of ROS and high cellular apoptotic ratios [128]. Therefore, the increased apoptotic activity coupled with ROS accumulation potentially heightens the selective pressure for malignant transformation.

HCC often exhibits the inactivation of tumor suppressor genes, which may be attributed to abnormal DNA epigenetic modifications, including methylation [129,130]. Benegiamo and colleagues [131] observed that in Huh-7 cells, HDV induces the expression of DNA methyltransferase 3b (DNMT3b) through STAT-3 activation. They also demonstrated that DNMT3b over-expression was associated with E2F1 transcription factor hypermethylation. In addition, using a cell cycle analysis, they showed that HDV induces G2/M arrest [131]. Another protein that has been involved in hepatocarcinogenesis is clusterin, which is over-produced in HCC [132]. In addition, the expression of clusterin was found to be substantially enhanced in metastatic HCC compared with primary tumors [133]. Using human hepatocellular carcinoma cell line Huh7, Liao et al. [134] demonstrated that both the L-HDAg and S- HDAg induced clusterin gene upregulation and that this event was associated with enhanced histone H3 acetylation within the clusterin promoter, thus suggesting that epigenetic changes induced by the HDV may contribute to the pathological outcome of HDV/HBV infection and HCC development.

Another potential epigenetic mechanism implicated in hepatocarcinogenesis induced by the HDV involves the modification of long non-coding RNA (lncRNA) expression. Specifically, the deregulation of lncRNA Y3 in HDV-related hepatocellular carcinoma (HCC) exemplifies how an HDV infection can influence the expression of lncRNAs, which play critical roles in gene regulation and cellular processes [135]. In this context, it has to be noted that the disruption in the regulation of lncRNAs plays a pivotal role in the replication process of the HDV [136].

Modifications in the cellular proteome have also been associated with an HDV infection. This has been highlighted by the altered expression of 89 proteins, mainly impacting DNA damage checkpoints and cell cycle regulation [137].

The first molecular profiling of HDV-related hepatocellular carcinoma (HCC) was provided in a study by Diaz et al. [138]. They analyzed liver specimens from patients with HDV-associated HCC and non-HCC HDV cirrhosis who underwent liver transplantation for HCC or end-stage liver disease. A significant finding was the downregulation of genes associated with hepatic fibrosis and the activation of hepatic stellate cells, potentially leading to the inhibition of extracellular matrix synthesis in the tumor microenvironment. More importantly, for the first time, they identified six pathways specifically associated with HDV–HCC: Hedgehog signaling, GADD45, DNA damage-induced 14-3-3σ, cyclins and cell cycle regulation, cell cycle G2/M DNA damage checkpoint regulation, and hereditary breast cancer (Figure 3). Notably, the majority of genes involved in these pathways were upregulated, implying that HDV–HCC leads to the enrichment of genes involved in DNA replication and/or DNA damage and repair, thereby substantiating the role of genomic instability in HDV-related HCC development. For further confirmation of the specificity of these pathways, Diaz et al. demonstrated that none of them were involved in HCC related to an HBV mono-infection, which was instead associated with metabolic processes, retinoic acid receptor signaling, cell remodeling, and motility functions. These findings underscore the fundamental differences between the molecular profiles of HDV–HCC and HBV–HCC.

A subsequent study by Yu and colleagues [139] employed microarray datasets to analyze cancerous and adjacent non-cancerous tissues from patients with CHB- or CHD-related HCC. They identified seven genes (CDC6, CDC45, CDCA5, CDCA8, CENPH, MCM4, and MCM7) primarily involved in the mitotic cell cycle and DNA replication. These genes were predominantly upregulated in the HCC subgroup related to CHD, revealing an HDV-selective impact on pathways involving these genes. Therefore, the study by Yu and colleagues underscores that the molecular signature of CHD-related HCC is characterized by an overexpression of genes critical for cell cycle progression and DNA replication/repair, further emphasizing genomic instability as a key mechanism in liver cancer development.

However, much of this data derives from limited-scale studies, underlining the importance of conducting a comprehensive analysis of the critical molecular characteristics of HCC linked to the HDV.

## 7. Clinical Features of HDV Infection and HCC Development

CHD is a severe liver disease characterized by a rapid progression to cirrhosis [2,3,80,140] and a higher rate of liver decompensation leading to death than a chronic HBV infection alone [83]. The course of CHD exhibits considerable variability and regional heterogeneity [141]. However, evidence indicates that at the time of diagnosis, between 30% and 70% of patients are found to already have cirrhosis. Moreover, within 5 to 10 years post-infection, cirrhosis develops in approximately 70% to 80% of cases [142]. Although over 50% of patients can have liver-related mortality within a decade [61], cirrhosis, once established, may persist as a stable disease for several years [143]. Individuals with an HDV infection have a markedly elevated risk of developing HCC compared to those with an HBV mono-infection. However, the specific role of the HDV in HCC development as well as the potential oncogenic nature of the virus is yet to be conclusively established [2]. This role is still controversial, because the HDV is a defective virus depending on the HBV, making it challenging to discern the specific contributions of both viruses in the oncogenic process [20,119].

Due to the lack of large prospective studies, it is also difficult to assess the incidence of long-term complications throughout the natural course of CHD, and the data available come prevalently from cross-sectional studies or retrospectives studies with long-term follow-ups. Furthermore, while certain studies have reported no statistically significant difference in the incidence rates of HCC between individuals with a dual HBV–HDV infection and those mono-infected with the HBV [6,144], numerous other studies have identified a higher risk of cancer in the former group compared to the latter [83,145,146,147]. Yet, complications related to portal hypertension were observed more frequently than the development of HCC [140,144,148].

HDV replication is associated with a significant risk of cirrhosis and HCC, with reported annual rates of 4% for cirrhosis and between 2.6% and 2.8% for HCC [83,149]. However, these figures are derived from studies focusing exclusively on patients co-infected with HBV–HDV, lacking a direct comparison with those solely infected with the HBV. In a critical contribution to the field, Fattovich et al. conducted a comparative study, demonstrating that the risks of HCC, liver decompensation, and mortality were, respectively, 3.2, 2.2 and 2-fold higher in anti-HDV-positive patients compared to those without an HDV infection in a cohort of patients with cirrhosis [147].

The persistence of HDV RNA positivity is strongly associated with both the progression to cirrhosis and an increased mortality rate [70,71,72]. Studies have also highlighted a direct correlation between elevated levels of HDV RNA in the bloodstream and a heightened risk of HCC development [73]. Additionally, HDV viremia appears to play an important role in promoting HCC development, even in patients whose HBV infection has been effectively suppressed by treatment with nucleotide/nucleoside analogues (NAs). This influence is underscored by data showing that the 5-year cumulative incidence of HCC was markedly different, standing at 7.3% for HDV-RNA-negative patients compared to 22.2% for those who were HDV-RNA positive, highlighting the substantial impact of HDV viremia on cancer risk even in the context of successful HBV suppression [150].

The evidence suggesting that the HDV may have a direct oncogenic role, independent of cirrhosis, is compelling, as demonstrated by two large cohort studies [151,152]. These studies showed that patients with an HDV infection have a six-fold higher risk of HCC development than those infected with the HBV alone, and that the incidence rate of HCC in HDV-positive individuals is 2.9 times greater than in HDV-negative ones [151,152]. Additionally, three systematic reviews and meta-analyses have highlighted the elevated risk of HCC in patients with CHD, ranging from 1.3 to 2.8 times higher than in those with an HBV mono-infection [12,153,154].

A critical meta-analysis by Alfaiate et al. [12], which pooled data from 25 cohort studies involving 75,427 patients and 68 case-control studies with 22,862 participants, found a significant increase in the HCC risk among CHD patients despite study heterogeneity (pooled odds ratio 1.28; 95% CI 1.05–1.57; I2 = 67.0%). These findings strongly support the link between an HDV infection and a higher risk of HCC development.

Furthermore, in a comprehensive meta-analysis by Chang et al. [153], subgroup analyses showed a significantly higher risk of HCC in patients with CHD compared to those with an HBV mono-infection. This increased risk was consistent across different ethnic groups and remained significant irrespective of a co-infection with HIV or HCV [153]. Supporting these findings, another detailed meta-analysis highlighted that despite the high prevalence of an HCV co-infection in patients with an HDV infection, the increased risk of HCC in individuals with the HDV was clearly demonstrated, even when cases co-infected with the HCV were excluded [12].

Abbas et al. [155] distinguished clinical and tumoral characteristics between HBV- and HDV-associated HCC. Indeed, HDV-associated HCC often presents with a smaller liver size, lower platelet count, and larger varices (indicating more severe portal hypertension) than HBV-related HCC, which is more likely to have multifocal tumors and higher alpha-fetoprotein levels [155].

A very recent multicenter Italian study analyzed the main clinical/oncological characteristics and the outcome of a large population of patients with HCC and positivity for anti-HDV antibodies from the Italian Liver Cancer database. HCC patients with an HBV/HDV co-infection had a worse liver function (*p* < 0.0001) than patients with the HBV, with a more frequent diagnosis of HCC during surveillance (*p* = 0.0001). Moreover, HBV/HDV patients had tumors with diffuse/infiltrating or massive behaviors (*p* = 0.023) less frequently and were more frequently classified as Milan-in (*p* = 0.005) than patients with the HBV alone. In this context, the Milan-in tumor stage (*p* < 0.0001) and surgical treatment, either liver transplantation (*p* < 0.0001) or liver resection (*p* = 0.044), were also confirmed as the only independent determinants of survival in HBV/HDV patients [11].

Contrasting data suggest that cirrhosis may be a more important risk factor for liver complications and HCC than the HDV infection itself. The study by Wranke et al. highlighted that while patients with CHD develop liver complications earlier (4.6 vs. 6.2 years) and more frequently (35.4% vs. 12.6%,) than those with an HBV mono-infection, CHB patients with cirrhosis have a higher incidence of HCC (35.5%) than CHD patients with cirrhosis (18.5%) [156].

In conclusion, there is considerable variability in disease progression in HDV-positive patients [157] due to differences in study methodologies, including the inclusion of patients based on anti-HDV antibody positivity without confirmation of HDV RNA presence. Geographical differences also contribute to this heterogeneity, reflecting differences in HDV genotypes, host genetic factors, and environmental exposures [141]. However, the advent of novel therapeutic options like BLV for patients with CHD heralds a new era in treatment. It will be important to await the results of extensive long-term research to determine whether these innovations are effective in reducing the incidence of HCC in this population.

## 8. Conclusions

CHD is a severe liver disease characterized by a rapid progression to cirrhosis and a higher risk of liver decompensation leading to death compared to a chronic HBV infection alone. Individuals with an HDV infection have a significantly elevated risk of developing HCC compared to those with an HBV mono-infection. However, the specific role of the HDV in HCC development as well as the potential oncogenic nature of the virus is still under debate. Several studies have demonstrated that an HDV infection induces a robust innate immune response, characterized by the activation of IFN signaling pathways and the production of ISGs. This innate immune response contributes to the pathogenesis of liver disease by promoting inflammation and liver cell damage. However, the adaptive immune response to the HDV is generally weak. The HDV has evolved several mechanisms to evade recognition by the adaptive immune system, including the selection of immune-escape variants and the suppression of innate immune cells such as NK and MAIT cells. As a consequence of the weak adaptive immune response, a chronic HDV infection is difficult to control and clear. One of the most critical aspects of an HDV infection is its association with an increased risk of HCC development. The molecular mechanisms underlying the involvement of the HDV in the development of HCC are not fully elucidated, but they are likely to involve both innate and adaptive immune responses, as well as various biomolecular mechanisms, including oxidative stress, epigenetic alterations, and the activation of cellular signaling pathways. Despite the challenges, several lines of consistent evidence have indicated that the risk of HCC occurrence can be reduced, although not eliminated, by current anti-viral treatments in patients with HBV. In conclusion, although new drugs such as BLV have emerged for patients with chronic HDV, it will be necessary to await new long-term studies to determine whether these treatments can effectively reduce the incidence of HCC in individuals with an HDV infection.

## Figures and Tables

**Figure 1 pathogens-13-00362-f001:**
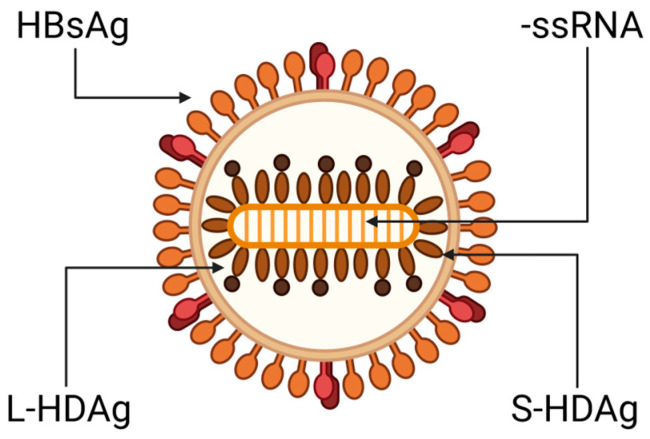
Schematic representation of HDV viral particle. Abbreviations: HBsAg (hepatitis B surface antigen); L-HDAg (large-hepatitis delta antigen); S-HDAg (small-hepatitis delta antigen); -ssRNA (negative single-stranded RNA). Created using BioRender.com.

**Figure 2 pathogens-13-00362-f002:**
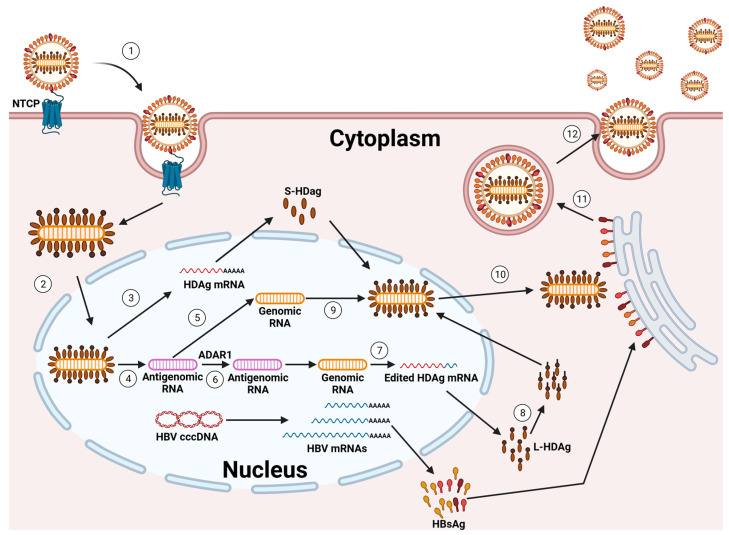
Life cycle of HDV. (1) The virus binds to the NTCP membrane receptor by the envelope composed of HBV HBsAgs. The viral particle then enters the cell through endocytosis, and the viral ribonucleoprotein is released into the cytoplasm. (2) The L- and S-HDAgs contain a nuclear localization signal that leads to the translocation of the viral ribonucleoprotein into the nucleus. (3) Here, the transcription of HDAg mRNA occurs via recruitment by the cellular RNA polymerase II. The HDAg mRNA is then exported to the cytoplasm, where it is translated to produce S-HDAgs. (4) During the first phase of replication, the HDV genomic RNA serves as a template to produce antigenomic RNA via RNA polymerase I. (5) The antigenomic RNA is then used by RNA polymerase II to produce new genomic RNAs. (6) The antigenomic RNA is also modified by the ADAR1 enzyme, which leads to the elimination of the stop codon of the S-HDAg. (7) The modified antigenomic RNA is replicated into the genomic RNA, thus inducing the transcription of the modified HDAg mRNA, which is exported to the cytoplasm, where this time, it leads to the production of the L-HDAg protein. (8) The L-HDAg contains a prenylation site that is farnesylated by a cellular farnesyltransferase before being translocated to the nucleus. (9) Both forms of the HDAg interact with the newly synthesized genomic RNA to form new viral ribonucleoproteins (RNPs) that are exported to the cytoplasm. (10) The L-HDAg, through its farnesylated cysteine, interacts with the cytosolic part of the HBsAg on the surface of the endoplasmic reticulum, thus inducing viral RNPs envelopment. (11) Enveloped viral particles are subsequently secreted through the endoplasmic reticulum (ER)–Golgi secretory pathway. (12) HDV virions exit the infected cell. The figure represents a cell infected with HBV, represented by the presence of cccDNA and the transcription of mRNA that lead to the translation of the HBsAgs necessary for the formation of the HDV envelope. Created using BioRender.com.

**Figure 3 pathogens-13-00362-f003:**
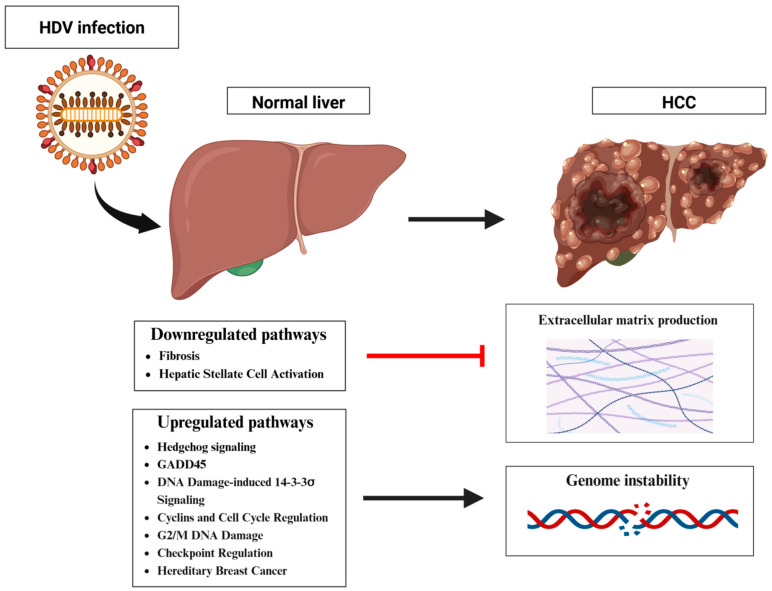
Schematic representation of the mechanisms by which HDV potentially induces HCC. The main downregulated pathways involved are as follows: hepatic fbrosis and hepatic stellate cell activation, while the most upregulated pathways are: Hedgehog signaling, GADD45, DNA damage-induced 14-3-3σ signaling, cyclins and cell cycle regulation, G2/M DNA damage, checkpoint regulation, and hereditary breast cancer. Created using BioRender.com.

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
