# Peer review of "Hepatitis Delta Virus and Hepatocellular Carcinoma"

_pathogens, 2024, doi:10.3390/pathogens13050362_

Round 1
Reviewer 1 Report
Comments and Suggestions for Authors
This seems a careful and up to date evaluation of the possible HDV roles in exacerbating the dangers of HBV infections.
I did not detect references to early studies of Chisari where mice were made transgenic for replicating HDV, and no pathogenesis was detected.
Similarly, there were no references to Chang who showed HDV alone caused cell cycle arrest.
Author Response
Point-by-Point Response to the Reviewer's Comments:
We thank the reviewer for his insightful feedback on our manuscript. We have carefully considered his comments and have incorporated them into the revised manuscript. The reviewer’s comments are shown in bold.
.
Reviewer 1:
- I did not detect references to early studies of Chisari where mice were made transgenic for replicating HDV, and no pathogenesis was detected.
Thank you for bringing this important reference to our attention. We apologize for the omission of Chisari's work in the original manuscript.
In the revised manuscript, we have incorporated the reference [113] and modified the text accordingly. Additionally, we have reviewed the literature and added several new references [114-118] to strengthen the discussion on HDV pathogenesis (Lines 407 - 419).
- Similarly, there were no references to Chang who showed HDV alone caused cell cycle arrest
We would like to express our gratitude for drawing our attention to the work by Chang. We appreciate your attention in conducting a thorough literature review.
Therefore, we have included two relevant studies in the manuscript. The first, by Hwang et al. [120] that demonstrates HDV-mediated cell cycle arrest. The second, by Wang et al. [121] that highlights the role of HDV in impairing cell proliferation.
Reviewer 2 Report
Comments and Suggestions for Authors
Dear Author,
Thanks for the paper entitled: Hepatitis Delta Virus and Hepatocellular Carcinoma. This is a review paper that describes HDV and hepatocellular carcinoma. In general, a good literature review was made including virology of HDV and pathogenesis of Hepatocellular carcinoma.
Minor issues:
Some references are missing in the epidemiology of HDV. In general, reviews were included, but primary studies could be also referenced. I suggest to include the following references about HDV epidemiology:
- https://pubmed.ncbi.nlm.nih.gov/29663457/
https://pubmed.ncbi.nlm.nih.gov/38030035/
https://pubmed.ncbi.nlm.nih.gov/37964693/
Author Response
Point-by-Point Response to the Reviewer's Comments:
We appreciate the reviewer's positive feedback and helpful suggestions for improving our study. The reviewer's comments are highlighted in bold.
Reviewer 2:
Minor issues:
Some references are missing in the epidemiology of HDV. In general, reviews were included, but primary studies could be also referenced. I suggest to include the following references about HDV epidemiology:
https://pubmed.ncbi.nlm.nih.gov/29663457/
https://pubmed.ncbi.nlm.nih.gov/38030035/
https://pubmed.ncbi.nlm.nih.gov/37964693/
We thank the reviewer for their valuable feedback on the epidemiology section. Now, we have incorporated the additional primary studies suggested to strengthen the discussion.
The revised manuscript includes the references provided by the reviewer (references 52-54).
Reviewer 3 Report
Comments and Suggestions for Authors
This is a worthwhile and comprehensive review of HDV and its role in HCC.
Minor Comments
Line 25: “associated with a rapid progression to cirrhosis and to high rates of development of hepatocellular carcinoma (HCC)”
- Should be “…cirrhosis and with high rates of…”
Line 101 :“Both genomic and antigenomic RNA is characterized by”
- Should be “are characterized by”
Line 119: “the following steps of HDV replication”
- Suggest “the subsequent steps” to improve clarity
Line 143: “which act as HDV secretion inhibitors”
- Should be “which acts as a HDV secretion inhibitor”
Line 240: “jdin more than 90%”
- Should this be “in more than 90%”?
Line 320: “While the innate response is activated in HDV infection, whereas the adaptive immune response is typically weak and unable to clear the virus consistently”
- Please reword this sentence as “while” then “whereas” is incorrect
Line 456: “suggesting that epigenetic changes induced by HDV may contributes to”
- Should be “may contribute to”
Figure 3
- (Downregulated pathways box): “Hepatic Fribrosis” – should be “Fibrosis”
- Figure legend “Sonic Hedgehof” should be “Hedgehog”
Line 513: “However, evidence indicates that the time of diagnosis”
- Should be “that at the time of diagnosis”
Line 624: “Despite the challenges, several consistent evidences have indicated”
- Please reword “several consistent evidences” as this is grammatically incorrect
Comments on the Quality of English LanguageMinor grammatical errors throughout.
Most have been highlighted under Minor Comments, but suggest final proof read by editors after corrections.
Author Response
Point-by-Point Response to the Reviewer comments:
We sincerely apologize for the oversights. We thank the reviewer for his careful review of the text. His edits and suggestions have greatly improved its clarity and comprehensiveness. The reviewer's comments are shown in bold.
Minor Comments
Line 25: “associated with a rapid progression to cirrhosis and to high rates of development of hepatocellular carcinoma (HCC)”
- Should be “…cirrhosis and with high rates of…”
As suggested, the sentence has been revised to read: '...cirrhosis and high rates of hepatocellular carcinoma (HCC) development...'
Line 101 :“Both genomic and antigenomic RNA is characterized by”
- Should be “are characterized by”
Line 101: The sentence has been corrected to: "Both genomic and antigenomic RNA are characterized by..."
Line 119: “the following steps of HDV replication”
- Suggest “the subsequent steps” to improve clarity
Line 119: We have adopted the reviewer’s suggestion and changed the wording to: "the subsequent steps of HDV replication" for improving clarity.
Line 143: “which act as HDV secretion inhibitors”
- Should be “which acts as a HDV secretion inhibitor”
Line 143: The sentence has been corrected to: "which acts as an HDV secretion inhibitor"
Line 240: “jdin more than 90%”
- Should this be “in more than 90%”?
Line 240: The sentence has been changed to: "in more than 90%"
Line 320: “While the innate response is activated in HDV infection, whereas the adaptive immune response is typically weak and unable to clear the virus consistently”
- Please reword this sentence as “while” then “whereas” is incorrect
Line 320: We acknowledge the incorrect use of "whereas" in this sentence. We have rephrased it as: "The innate immune response is activated during HDV infection, while the adaptive immune response is typically weak and unable to consistently clear the virus."
Line 456: “suggesting that epigenetic changes induced by HDV may contributes to”
- Should be “may contribute to”
Line 456: The grammatical error has been corrected. The sentence now reads: "suggesting that epigenetic changes induced by HDV may contribute to..."
Figure 3
- (Downregulated pathways box): “Hepatic Fribrosis” – should be “Fibrosis”
- Figure legend “Sonic Hedgehof” should be “Hedgehog”
Figure 3
- In the Downregulated pathways box, we have corrected "Hepatic fribrosis" to "Fibrosis".
- We have changed "Sonic Hedgehof" to "Hedgehog" in the figure legend.
Line 513: “However, evidence indicates that the time of diagnosis”
- Should be “that at the time of diagnosis”
Line 513:
We have added "at" to the sentence, changing it to "However, evidence indicates that at the time of diagnosis".
Line 624: “Despite the challenges, several consistent evidences have indicated”
- Please reword “several consistent evidences” as this is grammatically incorrect
Line 624:
We have rephrased "several consistent evidences" to "several lines of consistent evidence"